# TRPA1 Covalent Ligand JT010 Modifies T Lymphocyte Activation

**DOI:** 10.3390/biom14060632

**Published:** 2024-05-28

**Authors:** Katalin Szabó, Géza Makkai, János Konkoly, Viktória Kormos, Balázs Gaszner, Tímea Berki, Erika Pintér

**Affiliations:** 1Institute of Pharmacology and Pharmacotherapy, University of Pécs Medical School, H-7624 Pécs, Hungaryerika.pinter@aok.pte.hu (E.P.); 2Nano-Bio-Imaging Core Facility, University of Pécs Medical School, H-7624 Pécs, Hungary; 3Research Group for Mood Disorders, Department of Anatomy, University of Pécs Medical School, H-7624 Pécs, Hungary; 4Department of Immunology and Biotechnology, University of Pécs Clinical Center, H-7624 Pécs, Hungary

**Keywords:** TRPA1, lymphocytes, monocytes, CD4^+^ cells, CD14^+^ cells, B cells, dual RNAscope^®^ ISH-IF technique, TcR activation, JT010, intracellular Ca^2+^, flow cytometry

## Abstract

Transient Receptor Potential Ankyrin 1 (TRPA1) is a non-selective cation channel involved in sensitivity to a plethora of irritating agents and endogenous mediators of oxidative stress. TRPA1 influences neuroinflammation and macrophage and lymphocyte functions, but its role is controversial in immune cells. We reported earlier a detectable, but orders-of-magnitude-lower level of *Trpa1* mRNA in monocytes and lymphocytes than in sensory neurons by qRT-PCR analyses of cells from lymphoid organs of mice. Our present goals were to (a) further elucidate the expression of *Trpa1* mRNA in immune cells by RNAscope in situ hybridization (ISH) and (b) test the role of TRPA1 in lymphocyte activation. RNAscope ISH confirmed that *Trpa1* transcripts were detectable in CD14^+^ and CD4^+^ cells from the peritoneal cavity of mice. A selective TRPA1 agonist JT010 elevated Ca^2+^ levels in these cells only at high concentrations. However, a concentration-dependent inhibitory effect of JT010 was observed on T-cell receptor (TcR)-induced Ca^2+^ signals in CD4^+^ T lymphocytes, while JT010 neither modified B cell activation nor ionomycin-stimulated Ca^2+^ level. Based on our present and past findings, TRPA1 activation negatively modulates T lymphocyte activation, but it does not appear to be a key regulator of TcR-stimulated calcium signaling.

## 1. Introduction

Transient Receptor Potential Ankyrin 1 (TRPA1) is an ionotropic receptor [1] that mediates the neuro-immuno-epithelial interface network [2,3,4,5] in the airways [6,7], skin [8,9,10], and gut [3,11]. TRPA1 functions as a nonselective cation channel and chemosensor for sensitivity and sensitization to a plethora of irritating agents and endogenous mediators of oxidative stress and inflammation [1,12]. Its function has been proposed to contribute to a variety of interrelated sensory and inflammatory processes such as inflammatory hyperalgesia [13,14], colitis [15], and airway inflammation [16,17]. Absence, dysfunction, or inhibition of TRPA1 has been shown to decrease inflammation-related symptoms of psoriasis [9,10,18], rheumatoid arthritis [19], actinic keratosis [20], atopic dermatitis [21,22,23], and multiple sclerosis [24,25,26,27,28,29]. Selective TRPA1 antagonists were suggested as promising treatments for neuroinflammatory diseases [30,31]. However, discrepancies in the immune and nociceptive effects in preclinical and clinical studies highlighted that in addition to species-specific differences, the tissue and cellular environment also plays a crucial role in defining whether lack of TRPA1 function has a pro- or anti-inflammatory effect [31,32].

Expression of *Trpa1* mRNA in immune cells was reported based on comprehensive expression profile analysis of TRP channel gene families including TRPA1 in immune organs [33], by endpoint RT-PCR and qRT-PCR analysis, or by functional analysis in mouse immune cells, e.g., CD4^+^ splenocytes [34,35] and Th2 type T cells [32,36]. *Trpa1* transcription may also be regulated at the epigenetic level, as epigenetic alterations in pain-related syndromes and association analyses between chronic pain and the methylation pattern of TRPA1 suggested [37,38]. Understanding the role and functional properties of TRPA1 in immune cells is of particular importance for the design of potential therapeutic approaches, even if the target is TRPA1 on sensory neurons.

We reported earlier by qRT-PCR a detectable, but orders-of-magnitude-lower level of *Trpa1* mRNA in mouse monocytes and lymphocytes than in sensory neurons [39]. While controversial results were reported on TRPA1 protein and mRNA expression in monocytes or macrophages [39,40,41,42], a series of functional results suggest its presence and function in human monocytes and macrophages (reviewed in [12,32]). Functional patch clamp experiments, calcium signaling, and differences in cytokine secretion between the wild type (WT) and the TRPA1 knockout (KO) mice suggested the presence of a functional TRPA1 in CD4^+^ cells [34,35,39], though discrepancies in the conclusion based on the observed phenomena persist. Although TRPA1 was detected via Western blot in a Burkitt’s lymphoma cell line, no evidence was published on mRNA expression or function of TRPA1 in B lymphocytes.

This study intended to: (1) further elucidate the expression of *Trpa1* mRNA in immune cells by applying dual RNAscope^®^ in situ hybridization (ISH)–immunofluorescent (IF) technique, and (2) test the role of TRPA1 in lymphocyte activation using a synthetic covalent ligand of TRPA1, JT010, reported to be a selective potent agonist of the channel [43,44]. Matsubara et al. [45] provided evidence that JT010 is a much weaker TRPA1 agonist in mice dorsal root ganglion cells than in human fibroblast-like synoviocytes, at concentrations under 100 nM. They also reported that JT010 is a potent activator of human TRPA1 (hTRPA1), but not mouse TRPA1 (mTRPA1) expressed in human embryonic kidney (HEK) cells. However, since two critical cysteines, C621 (important for JT010 interaction) and C665 of hTRPA1, are conserved in mTRPA1 (C622 and C666) [46,47,48], we intended to test whether application of higher concentrations of JT010 modulates TRPA1-mediated Ca^2+^ influx in murine peritoneal cells. We aimed to test the effect of JT010 on mouse T cell function because of two reasons: (1) previous functional results indicated differences in calcium signaling and in cytokine secretion between WT and TRPA1 KO mice [34,39], and (2) in contrast to AITC and cinnamaldehyde, JT010 does not cause membrane damage of the primary immune cells even at high concentrations. In our earlier work [39], we could not determine the influence on TRPA1-mediated Ca^2+^ influx by other agonists in splenocytes and thymocytes, because these primary immune cells were vulnerable to these agents in our system, indicated by cell shrinkage, granulation, and compromised membrane permeability (shown by propidium iodide positive staining of the cells). We could not detect any significant differences in the TcR-induced Ca^2+^ signal of T lymphocytes isolated from spleen and thymus between the WT and pore-loop domain-deficient TRPA1 KO mice. However, imiquimod-stimulated elevation of Ca^2+^ level with sustained time kinetics was higher in the mouse TRPA1 KO CD8+ thymocytes [39].

The combined approach of RNAscope ISH and immunofluorescent staining is applicable in primary neurons and in tissue sections from rodent brain. It can be used for gene-specific detection in a specific cell population or for simultaneous identification of a target gene and protein in the same cells [49]. Other groups have applied a dual RNAscope ISH-IHC/IF technique with surface CD marker labeling, but independent protocols had to be developed for the particular studies [50,51,52,53]. Dual RNAscope/IHC combination is used in our laboratories for detection of intracellular proteins together with *Trpa1*, *Trpv1* mRNA expression [54,55,56,57,58]. Moreover, phenotyping of immune cells both for diagnostic or research purposes is performed routinely in our laboratories ([39,59,60,61], diagnostic laboratory at the Department of Immunology and Biotechnology, University of Pécs Clinical Center). Given our long-standing experience with these techniques, we studied the *Trpa1* mRNA expression in the context of CD marker expression by performing a quadruple IF-staining and RNAscope ISH.

## 2. Materials and Methods

### 2.1. Mice, Mononuclear Cell Isolation

Four-to-six-week-old male C57BL/6 mice were used in groups of two or three in experiments. Mice were kept and bred in the facility of the Institute of Immunology and Biotechnology, or that of the Institute of Pharmacology and Pharmacotherapy under conventional conditions. Thymuses were isolated and homogenized mechanically, peritoneal cells were washed out from the peritoneal cavity by RPMI supplemented with 5% fetal calf serum (RPMI/FCS) and used immediately for immunofluorescent staining at room temperature followed by either RNAscope ISH or measurement of intracellular calcium signaling.

### 2.2. Immunfluorescent Staining

For combined phenotype analysis and RNAscope ISH [62], 10^5^ cells/sample isolated from the peritoneal cavity were cultured in RPMI 1640 (R 6504, Sigma-Aldrich Chemie GmbH, Schnelldorf, Germany) supplemented with 10% fetal calf serum (FCS, A5209402, Gibco, Thermo Fisher Scientific, Waltham, MA, USA) on poly-D-lysin-coated (A 3890401, Sigma-Aldrich Chemie GmbH, Schnelldorf, Germany, diluted to 1 μg/mL final concentration in phosphate-buffered saline (PBS)) Superfrost Ultra Plus slides (Thermo Fisher Scientific, Waltham, MA, USA) in the presence of 5% CO_2_ for 3 h at 37 °C. After gentle washing with RPMI/FCS followed by PBS rinsing, cells were incubated in PBS containing 0.1% BSA (PBS/BSA) for 20 min at 4 °C. Cells were labeled with fluorochrome-conjugated cell surface antibodies at 4 °C in darkness for one hour, then washed with PBS/BSA 3 times for 20 min. The slides were incubated in 10% neutral buffered formalin solution (NBF, Cat. No.: HT501128, Merck KgaA, Rahway, NJ, USA) for 30 min at 4 °C.

For combined phenotype analysis and intracellular calcium signaling measurements, cells were cell-surface stained as described earlier [39,63] with some modification. To avoid, or at least reduce, the effects of cold temperature, mechanical stimuli, or PBS on TRPA1 and the internalization of the receptor, cells were washed and stained with RPMI/FCS instead of PBS/BSA. A total of 10^6^ cells were labeled with fluorochrome-conjugated cell surface antibodies for 30 min in darkness, then washed with RPMI/FCS prior to Ca^2+^ signal measurements on a FACSCalibur flow cytometer (Beckton Dickinson, Franklin Lakes, NJ, USA). Results were analyzed with FCS Express software 6 (De Novo Software, Pasadena, CA, USA)

We used appropriate combinations of the following antibodies purchased from BD Pharmingen, San Jose, CA, USA): rat anti-mouse CD4 Fluorescein (FITC, clone RM4-5, Cat. No.: 553046), rat anti-mouse CD4 Phycoerythrin (PE, clone H129.19, Cat. No.: 553653), rat anti-mouse CD8α PE-Cy5 (clone 53-6.7, Cat. No.: 553034); and the following antibodies purchased from Biolegends: rat anti-mouse CD14 PE (clone Sa14-2 Cat. No.: BZ-123310), Isotype Rat IgG2a, κ, (clone RTK2758, Cat. No.: BZ-400508). Rat mAb against B220 produced by anti-B220 hybridoma (clone RA3-6B2), purified by protein G chromatography and conjugated with FITC and Alexa Fluor 647 [60], was kindly provided by the Department of Immunology and Biotechnology, University of Pécs Clinical Center.

### 2.3. RNAscope In Situ Hybridization

In order to combine the surface marker staining on fixed immune cells with RNAscope ISH [62] for *Trpa1* mRNA detection, the pretreatment procedure was reduced to a protease III treatment (ACD, Cat. No.: 322381, 10 min, diluted 1:10 in PBS at RT). The pretreatment procedure was reduced to protease III (ACD, Cat. No.: 322381) treatment for 10 min (diluted tenfold in PBS) at RT. Further steps of the RNAscope protocol (probe hybridization, signal amplification and channel development) were performed according to the RNAscope Multiplex Fluorescent Reagent Kit v2 user’s manual (ACD, Hayward, CA, USA). Briefly, the surface marker-stained and formalin-fixed cells were treated with mouse *Trpa1* (ACD, Cat. No.: 400211) probes and then visualized using a TSA Vivid^TM^ Fluorophore kit 650 (Tocris, Cat. No.: 7527) (1:750) for the detection of *Trpa1* mRNA. A mouse 3-plex positive control probe (ACD; Cat. No.: 320881) specific to RNA polymerase II subunit A (*Polr2a*) mRNA, peptidyl-prolyl cis-trans isomerase B (*Ppib*) mRNA, ubiquitin C (*Ubc*) mRNA, and a 3-plex negative (ACD; Cat. No.: 320871) control probe for bacterial dihydrodipicolinate reductase (*dabP*) mRNA were tested on the samples as technical controls. After 2 × 15 min washes with PBS, the sections were counterstained with 4′,6-diamidino-2-phenylindole (DAPI) (ACD, Hayward, CA, USA). Finally, after a PBS wash, the slides were covered with 50% glycerol dissolved in PBS mounting medium. Slides were stored at −20 °C until confocal microscopy.

### 2.4. Confocal Microscopy, Semi-Quantitative Image Segmentation, and Co-Localization Analysis

Fluorescent labeling was imaged utilizing either using an Olympus FluoView 1000 confocal microscope (Olympus, Hamburg, Germany) or Nikon Ti-2 C2 confocal microscope (Nikon Europe, Amsterdam, The Netherlands). Digital images were captured by sequentially scanning in analogue mode for the respective fluorophores in order to avoid false positive signals due to the slightly overlapping emission spectra, and to detect reliably quantifiable fluorescent signals. Cells labeled with combined immunofluorescent and RNAscope ISH were scanned across four channels corresponding to their respective wavelengths. The resulting digital images from the four channels, representing the same area, were automatically overlaid and merged. Image segmentation and co-localization measurements were performed using the following freely available software: Fiji ImageJ (version 1.54f) [64] and CellProfiler (version 4.2.6) [65], employing 4 to 16 unedited images from each corresponding channel.

### 2.5. Measurement of Intracellular Calcium Signaling by Flow Cytometry

Intracellular free Ca^2+^ was measured using Fluo-3-AM (Molecular Probes, Cat. No.: F-1242 [66]) according to the protocol described earlier [39,67,68]. Fluo-3-AM 1 mg/mL stock solution was dissolved in Pluronic-F-127 (Sigma P 2443) and additional dimethyl sulfoxide (DMSO). Briefly, isolated cells were suspended in RPMI supplemented with 5% FBS and 2 mM CaCl_2_ (CaRPMI). Cells (2 × 10^6^ cells/mL) were stained with fluorochrome-conjugated cell surface antibodies as described above in Section 2.2, then incubated with Fluo-3-AM in a CO_2_ incubator for 30 min at 37 °C. Cell suspension was washed and incubated for an additional 20 min in CaRPMI, then fluorescence was measured immediately using a Becton Dickinson FacsCalibur flow cytometer (Franklin Lakes, NJ, USA). Mean fluorescence intensity of Fluo-3 dye (FL1) was monitored before and after stimulation for 10 min in CD4^+^, CD8^+^, B220^+^, and CD14^+^ subpopulations and in size- and granulation-defined subpopulations based on FSC/SSC values.

To test effects of JT010 (6269, Tocris Bioscience, Bristol, UK) on intracellular Ca^2+^ level of the cells, JT010 or DMSO as control were added either alone, simultaneously with anti-CD3, or also by pre-incubating the cells with JT010 prior to addition of anti-CD3 and crosslinking agent IgG. Identical volumes (0.15% *v*/*v*) of concentration series of JT010 stock dilutions dissolved in DMSO in glass vials were used to achieve the proper final concentration in CaRPMI, to avoid JT010 absorption to polypropylene surfaces described by Heber 2019 [69].

For TcR-dependent Ca^2+^ signal analyses, cells were stimulated with hamster anti-mouse CD3 Epsilon monoclonal antibody (Clone 145-2C11, Cat. No.: MAB484, RD systems, Minneapolis, MN, USA) for 10 min, then crosslinked with goat anti-hamster IgG (Abcam, Cat. No.: ab5738) as described earlier [39,67,68]. In the case of B cell activation analyses, the baseline Fluo-3 fluorescence was measured for 1 min, then cells were stimulated with anti-mouse-IgG [68]. Ionomycin (2 μM, I 0634, Sigma-Aldrich Chemie GmbH, Schnelldorf, Germany) was applied at the end of every experiment for internal control to check nonresponsive cells and acquire a maximum response. Fluorescence was measured on a FACSCalibur flow cytometer (Beckton Dickinson, Franklin Lakes, NJ, USA). Data were analyzed by the Cell Quest software program (version 3.1, BD Biosciences, San Jose, CA, USA). Changes in Ca^2+^-indicator fluorescence were calculated as a ratio to that of non-stimulated quiescent cells.

### 2.6. Monitoring Cell Death and Plasma Membrane Damage through Measuring Phosphatidylserine (PS) Exposure of the Cells by Annexin V–Binding Assay

Cell death and plasma membrane damage of the cells were monitored by analysis of PS exposure of cells by annexin V–binding assay as described by Albrecht et al. [70] with minor modifications: instead of propidium iodide, 7-AminoactinomycinD (7-AAD, A 9400, Sigma-Aldrich Chemie GmbH, Schnelldorf, Germany) was used. Briefly, before addition of JT010, cells were preincubated in annexin V–binding buffer for 2 min prior to addition of 7-AAD and Alexa Fluor 488-conjugated annexin V (1:80 *v*/*v*; Molecular Probes, Eugene, OR, USA). After the addition of either JT010 or DMSO (as control), changes in annexin V–binding (1:80 *v*/*v*; Molecular Probes, Eugene, OR, USA) and 7-AAD over time were recorded for 10 min. In addition, cell death and plasma membrane damage of the cells were monitored also during simultaneous addition of JT010 and anti-CD3 and crosslinking agent IgG, and after preincubation with JT010 followed by anti-CD3 and crosslinking agent IgG. Ionomycin (2 μM, I 0634, Sigma-Aldrich Chemie GmbH, Schnelldorf, Germany) was applied at the end of every experiment for internal control to check nonresponsive cells and acquire a maximum response of PS exposure. Fluorescence was measured on a FACSCalibur flow cytometer (Beckton Dickinson, Franklin Lakes, NJ, USA) flow cytometer. Data were analyzed by the Cell Quest software program (version 3.1, BD Biosciences, San Jose, CA, USA).

### 2.7. Statistical Methods

Experiments were performed at least 3 independent times, analyzing the effects on at least 3 mice. Results were analyzed with GraphPad Prism 8.0 software (La Jolla, CA, USA). Data were expressed as mean ± standard error (SEM) of the values received in independent experiments. Statistical evaluation was performed using Student’s *t*-tests, where *p*-values < 0.05 were considered significant.

## 3. Results

### 3.1. Trpa1 Transcripts Were Detected in CD4^+^ and CD14^+^ Cells by RNA Scope In Situ Hybridization

To evaluate which immune cells express *Trpa1* mRNA, we developed a protocol to perform a quadruple staining by combining the RNAscope^®^ ISH technology with the classical immunofluorescent staining (IF). We aimed to preserve the morphology of the cells, reduce the digestion of the cell surface epitopes by protease pretreatment procedure of the RNAscope ISH technique, and reduce the internalization of CD markers, testing a series of cell surface staining and cell fixation methods and varying the sequence of the IF staining and RNAscope ISH.

#### 3.1.1. Trpa1 mRNA Was Detected by RNAscope ISH in CD4^+^ Cells

Confocal laser scanning microscopy revealed that *Trpa1* mRNA is expressed in anti-CD4 antibody labeled cells isolated from peritoneal cavity of mice. Co-localization of *Trpa1* with anti-B220 antibody-stained cells was not detected, as the representative image series of the quadruple staining shows in Figure 1a. *Trpa1* transcripts were also present in solely DAPI-stained cells, which may indicate either expression of *Trpa1* in the CD4^−^/B220^−^ mononuclear cell types in the peritoneal cavity, or loss of epitope-bound fluorescent antibodies during the RNAscope procedure. As the bright-field superimposition (merge) of the same field of this image shows (Figure 1b), the anti-CD4^+^ labeling can be seen both on solitary cells and at the contact point of two interacting cells, where the labeling was presumably more readily preserved despite protease treatment. It is worth noting that our protocol preserved the morphology of the cells, as the immunostaining on viable, non-permeabilized cells was carried out prior to RNAscope procedure. In our final protocol, we used direct IF labeling instead of indirect IF labeling to avoid non-specific binding of the secondary antibody after or during the RNAscope procedure. The disadvantage of the method was the much weaker signal, while the benefit was a more specific staining, as seen detailed in Appendix A and in Figure 2.

#### 3.1.2. Trpa1 Transcripts Were Detected in CD14^+^, CD4^+^ Cells but Not in B Cells

The quadruple staining allowed us to visualize two CD marker proteins together with the cell nuclei (DAPI) and the RNAscope signals on the same cell subsets. *Trpa1* mRNA was detected in CD14^+^ and CD4^+^ cells, but not in B220^+^ cells, as demonstrated in Figure 2 and Appendix A. *Trpa1* transcripts were localized inside CD14^+^ cells and CD4^+^ cells, while red labeling was absent in isotype control antibody-stained samples (IT CTR), representing the specificity of the staining with these antibodies. A mouse 3-plex positive control probe specific to RNA polymerase II subunit A mRNA (+RNA scope CTR), and a 3-plex negative control probe (-RNA scope CTR) to bacterial dihydrodipicolinate reductase mRNA validated the modified RNAscope method. The same anti-CD14 antibody was successfully used for immunohistochemistry and flow cytometric analysis to label isolated cells from the spinal cord and from the peritoneal cavity [71,72]. B220 antigens represent a subset of mouse CD45 isoforms predominantly expressed on all B lymphocytes, including pro-, mature, and activated B cells [60]. The antibody used for CD4 antigen detection was successfully used earlier in phenotyping T cell subpopulations of different sources [39,60,61]. Partial permeabilization of the cells by 10% NBF containing approx. 4% formaldehyde or internalization of CD markers before fixation may explain why the CD marker’s immunofluorescent signals appeared not only at the plasma membrane but also intracellularly.

Appendix A shows a lower magnification of the same test series, presenting more clearly the abundance of *Trpa1* transcripts and the presence of all three phenotypes in the peritoneal cavity samples. It is important to note here that the frequency of the B220^+^, CD14^+^, and CD4^+^ cell types in these images does not reflect the distribution of these cell phenotypes in the peritoneal cavity of the mice. The ratio of these cells was modified by the differences in cell capability of attachment, their vulnerability, the epitope, and the antibody sensitivity to protease treatment followed by the RNAscope procedure (see Figure 4a in Section 3.2 for CD marker surface staining of viable peritoneal cells by flow cytometry). While specificity can be seen clearly in the case of the more faintly stained cells, we could not avoid some harsh dual- or triple-labeled artifacts by any tried modification of the method without losing the plasma membrane-specific staining of the cells. As Appendix A demonstrates by the superimposition of the bright field of the images, the cells with this harsh dual or triple labeling of the cells with RNAscope ISH and either one or two of the antibodies were the particular cells that appeared darker in the bright-field view. The exclusion of this type as artifacts was crucial for semi-quantification of the co-localization results, since these were presumably cells in which cell death processes (necrotic or apoptotic) had occurred, resulting in non-specific sticking of the antibodies.

#### 3.1.3. Semi-Quantification of mRNA Expression and CD Marker Co-Localization

Figure 3 summarizes *Trpa1* mRNA expression in CD marker-labeled peritoneal cells visualized by fluorescence microscopy. The number of cells double-labeled with both RNAscope signals and the particular CD marker antibody was counted, as well as the total cell number based on the DAPI-stained nuclei in the same image. The ratio of the double-labeled cells to the total cell number in the image are expressed as the percentage of mRNA-expressing cells of CD4^+^, CD14^+^, B220^+^ or non-CD marker-labeled −/− cells (such as CD4^−^/B220^−^ or CD14^−^/B220^−^ cells). *Trpa1* mRNA expression was compared to RNAscope + control and – control signals of each phenotyped cells; mean +/− SEM values are shown for each phenotype separately in Figure 3. Statistically significant differences were signed between the values of the *Trpa1* mRNA or the RNAscope + control co-localization compared to co-localization of the RNAscope–control in CD4^+^, CD14^+^, B220^+^, and −/− cells. The significant differences reflect the *Trpa1*-specific transcript abundance or *Polr2a* compared to any potential non-specific background amplification of RNAscope–control bacterial *dabP* probes.

As Figure 3 shows, specific surface immunofluorescent staining with CD markers of viable peritoneal cells combined with modified RNAscope ISH technique evaluated that *Trpa1* mRNA is expressed in CD14^+^ and CD4^+^ cells and also in non-CD marker-labeled cells, but mRNA expression of *Trpa1* was not significant in B cells. The relative mRNA levels can only be compared to the + and the − RNA scope control signal levels in the particular CD marker-labeled cells by this technique. Despite these limitations, the established dual RNAscope-IF assay was useful to detect small amount of *Trpa1* transcripts, when testing the cells immediately after isolation from the mice without enrichment, sorting or separation methods, with minimal intervention of cell activation or stage.

We used the CellProfiler program (v4.2.6) for quantification of the quadruple-labeled cell images following the steps as presented in Appendix A. To ensure that Cellprofiler image analysis worked properly on our images, we first standardized the workflow pipeline using the RNAscope positive and negative control images for combined RNAscope and IF. As demonstrated in Appendix A, using the dual RNAscope^®^-IF technique, we could not avoid cell damage even though we started the procedure with 90 +/− 5% viable cells. Therefore, the quantification needed careful evaluation of artifacts caused by the technical challenges of labeling plasma membrane-localized proteins with RNAscope^®^ ISH method (see Appendix A).

As noted already in Section 3.1.2, the values of *Trpa1*-expressing cells cannot be compared between the different phenotypes of cells in the peritoneal cavity, since the frequency of the cell types in these images is modified by the differences in cell capability of attachment, vulnerability, the epitope, and antibody sensitivity to protease treatment followed by RNAscope procedure. Taking this also into account, our result confirmed that a significant number of cells expressed *Trpa1* mRNA in CD4^+^ cells, in CD14^+^ cells, and in non-CD marker-labeled (−/−) cells in the peritoneal cavity of mice, with each frequency comparable to *Polr2a* mRNA transcript frequency in the same cell type.

### 3.2. JT010 Stimulated Significant Elevation of Intracellular Ca^2+^ Level of Peritoneal and Thymus Cells Only at High Concentration

To evaluate whether in vitro stimulation of peritoneal cells (PEC) or thymocytes by the potent selective covalent TRPA1 agonist, JT010 [43,48], modulates intracellular Ca^2+^ levels of mononuclear immune cells; changes in Ca^2+^ levels in cells were assessed by monitoring the fluorescence of the Ca^2+^ indicator Fluo-3 by flow cytometry.

As flow cytometry analyses show in Figure 4a, cells isolated from the peritoneal cavity were identified as CD4^+^ (middle left panel) or B220^+^ lymphocytes (bottom left panel) or CD14^+^ mononuclear cells (middle right panel). Lymphocytes were detected in gate G1 based on FSC/SSC parameter distribution, mostly as CD4^+^ (27% +/− 5) or B220^+^ (37% +/− 5) lymphocytes; less than 7% of the peritoneal cells in this G1 gate were CD8+ cells. The high ratio of the double negative peritoneal cells (54% +/− 13) by flow cytometry indicated that the similarly high ratio of non-CD marker antibody-stained cells detected by RNAscope^®^ ISH-IF technique (Figure 3) were not the consequence of the protease treatment or reduction of fluorescence by the procedure of the RNAscope^®^ ISH-IF technique. As Figure 4a shows, the cells in gate G3 and gate G4 proved to be CD14^+^/CD4^-^ cells, as did close to 100% of the cells in gate G3, and 10% +/− 3 of the cells in gate G2.

A significant sustained elevation of intracellular free Ca^2+^ levels could be detected only at high concentration of JT010 (40 μM) in mouse PEC lymphocytes (Figure 4b), PEC CD4^+^ lymphocytes (Figure 4c), and thymus lymphocytes (Figure 4d). We tested TcR-activated Ca^2+^ signals (CD3) to ensure lymphocyte responsiveness and the effect of DMSO as vehicle control, and ionomycin was applied at the end of each experiment for internal control to check nonresponsive cells and acquire a maximum response. The much weaker potency of synthetic compound JT010 to activate mouse TRPA1 than human TRPA1 was reported by Matsubara et al. [45], reflecting a much lower affinity of JT010 toward mouse TRPA1 than for human TRPA1.

To ensure the integrity of the cells, PS exposure of the plasma membrane was monitored by annexin V–binding assay and the simultaneous presence of 7-AAD in the samples. As Figure 4e demonstrates, no significant changes of PS exposure of cells have been observed during the measurements over time in PEC lymphocytes; only the addition of ionomycin at the end of the experiments increased Annexin V-binding, as was expected (1–2 μM of calcium ionophore ionomycin causes calcium influx and PS exposure of lymphocytes [70]).

### 3.3. JT010 Inhibited TcR-Induced the Ca^2+^ Signal of Lymphocytes and CD4^+^ Lymphocytes a Concentration-Dependent Manner

Since addition of JT010 covalent ligand alone caused only sustained elevation of intracellular Ca^2+^ levels, in contrast to the fast and transient TcR-activated Ca^2+^ signals of the same cell types (see reviews on kinetic properties of Ca^2+^-signals [73,74]), our main question remained to be answered: whether the function of the TRPA1 channel contributes to the activation of the T lymphocytes.

To address this question, we tested whether JT010 treatment at a concentration range between 1–40 μM modifies the T-cell receptor-stimulated Ca^2+^ signal of peritoneal lymphocytes. As seen in Figure 5, concentration-dependent inhibitory effects of JT010 were observed on the TcR-induced Ca^2+^ signal, significant above 4 μM in the case of PEC lymphocytes (Figure 5a,c) and above 2 μM in the CD4^+^ lymphocytes (Figure 5b,d). The same volume of vehicle control DMSO addition did not decrease TcR-stimulated Ca^2+^ signals of the cells. However, as the non-significant slight difference between the relative MFI values of Ca^2+^ signal in the case of DMSO (0.15% *v*/*v*) and that of the solely anti-CD3 stimulated Ca^2+^ signal indicated (see in Figure 5b,d at zero JT010 concentration of x scale), even DMSO itself elevated slightly MFI values of the Ca^2+^ signal.

As detailed in Figure 4a, we gated PEC lymphocytes based on FSC/SSC parameter distribution; in these cases, the T lymphocyte activation was distinguished based on the specific nature of the activation, since anti-CD3 cross-linking activates T cells selectively (see detailed method in Section 2.5). No difference in the inhibitory effects was observed dependent on whether co-addition of JT010 with anti-CD3 antibody happened or anti-CD3 cross-linking was preceded by JT010 pre-incubation of the cells.

To test whether the inhibitory effect of 40 μM JT010 on TcR-stimulated Ca^2+^ signals was not due to compromised plasma membrane properties, we followed fluorescent annexin V–binding to the cell surface (Figure 5e upper panel) parallel with 7-AAD staining of cells (Figure 5e middle panel) over time detected by flow cytometry. Cell surface bound Annexin V fluorescence is proportional to the PS exposure in the outer membrane leaflet. 7-AAD, similarly to propidium iodide, stains cell nuclei (DNA) only in the case if capable of entering the cells. As the representative plots in Figure 5e demonstrate, 40 μM of JT010 treatment did not cause elevation of the Annexin V labeling or the 7-AAD fluorescence of PEC lymphocytes; only addition of ionomycin (2 μM) at the end of the experiment resulted in maximum PS exposure of cells. Taken together, JT010 administration, even at high concentrations, did not modify cell size and granulation, 7-AAD staining or Annexin V-binding, as had been detected and reported earlier by other agonists (AITC, cinnamaldehyde) and antagonists (HC-030031 and A967079) [39].

### 3.4. JT010 Neither Modified Peritoneal B Cell Activation nor Ionophore Ionomycin-Stimulated Elevation of Intracellular Ca^2+^ Level of Cells Isolated from Peritoneal Cavity of Mice

To probe whether this inhibitory effect of JT010 is restricted to T cell activation, we tested the Ca^2+^ signal activated by anti-IgG antibody through the B cell receptor (BcR) of PEC lymphocytes and that of the B220^+^ lymphocytes.

No significant inhibition on rapid Ca^2+^ signal characteristic on IgG-activated response [68,75] was detected at a concentration range between 1–40 μM of JT010 that modified T-cell receptor-stimulated Ca^2+^ signals of PEC lymphocytes (Figure 6a) and B220^+^ lymphocytes (Figure 6b). Significantly, a lower intracellular Ca^2+^ level occurred only at later time-points of measurements (460 or 540 s) by 40 μM JT010 pre-incubation compared to either the vehicle control DMSO (at 460 s: 1.06 +/− 0.14 versus 2.10 +/− 0.28, mean fold of baseline MFI +/− SEM, n = 3, *p* ≤ 0.05) or compared to the non-pre-incubated IgG-induced Ca^2+^ level (at 460 s: 1.06 +/− 0.14 versus 1.98 +/− 0.2, mean fold of baseline MFI +/− SEM, n = 4, *p* ≤ 0.05) in the case of PEC lymphocytes. Similarly, lower intracellular Ca^2+^ level was observed at this late time of detection in B220^+^ lymphocytes: 40 μM JT010 pre-incubation compared to the vehicle control DMSO (at 460 s: 1.11 +/− 0.12 versus 2.35 +/− 0.31, mean fold of baseline MFI +/− SEM, n = 3, *p* ≤ 0.05), or compared to the non-pre-incubated IgG-induced Ca^2+^ level (at 460 s: 1.11 +/− 0.12 versus 2.39 +/− 0.22, mean fold of baseline MFI +/− SEM, n = 4, *p* ≤ 0.05).

As shown detailed in Figure 4a, we gated lymphocytes with high B220^+^ expression level (bottom left panel in Figure 4a), reflecting the B-2 subpopulation of the mouse peritoneal cavity B lymphocytes [62,76,77,78,79,80].

In the case of “PEC lymphocytes” where gating was based on FSC/SSC parameter distribution, the BcR-dependent lymphocyte activation was distinguished based on the specific nature of the activation, since anti-IgG antibody activates B lymphocytes selectively (see detailed method in Section 2.5 [68]).

We further tested concentration-dependent perturbance of the intracellular Ca^2+^ level of lymphocytes by investigating ionomycin-stimulated activation of lymphocytes. JT010 pre-incubation did not result in significant inhibition of 2 μM ionomycin-induced Ca^2+^ levels of PEC lymphocytes (Figure 6b) or CD4^+^ lymphocytes at the concentration that modified T-cell receptor-stimulated Ca^2+^ signal of these cells.

Taken together, these suggest that the surprising inhibitory effect of JT010 is specific to activation of PEC T lymphocytes and CD4^+^ PEC lymphocytes.

## 4. Discussion

In this report, we provided evidence that *Trpa1* is expressed in mouse peritoneal CD4^+^ lymphocytes and revealed a T lymphocyte-specific concentration-dependent effect of the selective covalent TRPA1 agonist, JT010, following treatment of peritoneal cells [43]. JT010 inhibited TcR-mediated activation of CD4^+^ peritoneal lymphocytes in a concentration-dependent manner, but did not block peritoneal B lymphocyte activation, which suggests a potential regulatory role of TRPA1 in peritoneal T cell activation. The peritoneal cavity contains the bulk of the body’s immune cells, including macrophages, mast cells, T cells, B cells, eosinophils, neutrophils, natural killer cells, and dendritic cells [81,82].

*Trpa1* mRNA was detected by the dual RNAscope^®^ ISH-IF technique, capable of detecting a single mRNA transcript, including partially degraded mRNA, combined with the cell surface staining of living CD4^+^ and B220^+^ lymphocytes and CD14^+^ cells. By using this powerful technique, we extended the findings by methods used in previous [34,35] analyses of *Trpa1* expression, including ours [39]. The RNAscope^®^ ISH-IF can detect small amounts of *Trpa1* transcripts in a morphological context of the cells, without enrichment or sorting or separation methods, and with minimal intervention of cell activation or stage. Although the relative *Trpa1* mRNA levels in CD4^+^ and B220^+^ lymphocytes, CD14^+^ cells, and non-phenotyped cells can only be compared to the RNAscope positive and negative control signal levels in the particular CD marker-labeled cells, the semi-quantification of *Trpa1* transcripts was possible by using this method. The present results are consistent with our earlier results by qRT-PCR analyses, where we also detected low levels of *Trpa1* mRNA in monocytes and lymphocytes isolated from primary and secondary lymphoid organs of mice [39]. While the expression of *Trpa1* mRNA was reported in mouse CD4^+^ splenocytes and thymocytes either by RT-PCR or qRT-PCR analyses [34,35,39], other studies did not support transcription of *Trpa1* in lymphocytes or in peritoneal macrophages [83,84,85]. The differences in these findings may be explained by the methodical differences. In fact, we intended to use CD14^+^ cells as a positive control in optimizing our dual RNAscope^®^ ISH-IF protocol, since an increasing amount of evidence suggests that TRPA1 receptor/channel functions in mouse and human macrophages (reviewed in [32]). We detected *Trpa1* transcripts co-localized with anti-CD14 antibody labeling in the peritoneal cell samples, which suggests a monocyte-derived subpopulation of cells in the peritoneal cavity that express *Trpa1*. Further research would be needed to identify the exact phenotype of these cells. Our finding that *Trpa1* mRNA was not found in cells characterized by high B220^+^ antigen staining suggests that B2 lymphocytes in this niche do not express *Trpa1.* This is consistent with the absence of the published evidence of mRNA expression or TRPA1 function in B lymphocytes [32].

Although *Trpa1* mRNA levels were orders of magnitude lower in lymphocytes and macrophages than in trigeminal ganglion cells [39], we provided evidence of *Trpa1* mRNA expression in CD4^+^ lymphocytes by two methods, earlier by qRT-PCR and presently by the RNAscope technique.

In our earlier study, we analysed the immune phenotype and activation characteristics of the pore-loop domain-deficient TRPA1 KO mice (Ca^2+^ channel functional KO) and found no difference in TcR activation between WT and KO mutant lymphocytes prepared from spleen and thymus [39], contrary to results by Bertin et al. [34]. In our previous work, we aimed to analyze the effects of agonists, such as AITC and cinnamaldehyde, or antagonists, such as HC-030031 and A967079 on T cell activation, but our efforts, contrary to those of other groups (Bertin et al. [34], Sahoo et al. [35]), failed to demonstrate TRPA1-dependent effects due to the observed cell shrinkage, granulation, and membrane damage parallel with changes in intracellular Ca^2+^ level. In our present work, we instead studied the effects of a selective covalent agonist, JT010, on T cell activation.

The concentration-dependent inhibitory effect of JT010 on TcR-induced Ca^2+^ signals in peritoneal lymphocytes suggests a covalent TRPA1-ligand-dependent effect in our present study. The addition of JT010 did not compromise the integrity of the lymphocytes or their plasma membrane, as shown by the analysis of PS exposure of cells measured by annexin V–binding assay. This is further confirmed by the findings that JT010 neither modified peritoneal B cell activation nor ionophore ionomycin-stimulated elevation of intracellular Ca^2+^ levels. In agreement with Matsubara et al. [45], who reported a large difference in JT010 capability to activate mTRPA1 versus hTRPA1, we found that only high JT010 concentrations were effective in stimulating a significant sustained elevation of the intracellular Ca^2+^ levels and in inhibiting the TcR-mediated Ca^2+^ signal in mouse lymphocytes. The half maximum concentration required for hTRPA1 activation is reported to be 0.65 nM in calcium uptake assays and 7.6 nM in electrophysiological studies [43,48]. Matsubara et al. estimated the half maximum concentration for hTRPA1 activation to be 3–10 nM, while application of higher JT010 concentrations (at maximum 1000 nM) induced a moderate Ca^2+^ response to mTRPA1 activation, suggesting that mTRPA1 is much less sensitive to JT010 than hTRPA1. The highest concentration of JT010 applied in electrophysiological experiments was 100 nM of JT010. The two critical cysteines, C621 (important for JT010 interaction) and C665, of hTRPA1 are conserved in mTRPA1 (C622 and C666) [46,47,48], Matsubara et al. identified non-conserved residues in hTRPA1 and mTRPA1 that influenced the response to JT010 by measuring the ratio of the peak Ca^2+^ response of mutant TRPA1 versions compared to control HEK cells and in electrophysiological experiments. They found that even if the different residues of mTRPA1 were substituted with cysteine of hTRPA1, these mutants showed low sensitivity to 100 nM JT010 in patch clamp experiments. This suggests that the much weaker potency of JT010 to activate TRPA1 reflects a more complex determination of TRPA1 interaction with JT010.

Several TRP channels have been reported as important regulators of TcR signaling [74], by regulating the activity of store-operated Ca^2+^ signaling (SOCE), which is dependent on cell membrane depolarization [86]. TRPM4, a Ca^2+^-activated Na^+^-selective channel in T cells, serves as one of the negative feedback mechanisms during T cell activation upon SOCE [87,88]. Ca^2+^-induced activation of TRPM4 triggers Na^+^ influx through the channel; it depolarizes the plasma membrane and restricts further Ca^2+^ influx through ORA1. The Mg^2+^- and Ca^2+^-permeable TRPM7, by its kinase domain activity, regulates ORAI1 signaling and coordinates antigen receptor signaling termination in B lymphocytes, likely through phosphorylation of phospholipase Cγ (PLCγ) isoforms1 [89]. Knockdown of TRPC3 caused a small reduction in SOCE [90]; functioning of SOCE in human platelets required the association of STIM1, the Ca^2+^-permeable channel Orai1, TRPC1 and TRPC6 [91,92,93]. A potential mechanism for TRPA1-dependent modulation of SOCE was also proposed in megakaryocytes, a protein–protein binding transmitted association of TRPA1 with STIM1, Orai1, TRPC1, and TRPC6, based on the results of thrombin-stimulated co-immunoprecipitation of these proteins [94]. One of the characteristic features of TRPA1 is the high number of ankyrin repeats at the cytoplasmic NH2-terminal tail, which might provide the protein elasticity, as well as the ability to interact with other proteins [1]. The interactions of the TRPA1 channel with other proteins is an important research area, most probably in terms of regulation of both function and intracellular localization, trafficking, and internalization of TRPA1 [1]. The best-studied example so far is that of TRPA1 and TRPV1, where coexpression of the channels results in distinct activation profiles from those of cells expressing only TRPA1 or TRPV1 [95,96,97]. In neuronal cells, TRPA1 is generally co-expressed with TRPV1, and in vitro, direct evidence was reported that formation of a channel by two molecules of TRPA1 together with two monomers of TRPV1 is possible [98]. A mutation in TRPV1 influenced the voltage dependency, Ca^2+^ sensitivity, and magnitude of TRPA1-mediated currents [99]. The Ca^2+^-triggered activation of TRPA1 is attenuated by TRPV1 in the presence of extracellular Ca^2+^, but not in Ca^2+^-free conditions, suggesting that this modulation may work rather as a direct protein–protein interaction thsn through TRPV1 providing Ca^2+^ microdomains for TRPA1 activation. The interaction may also act through another coupling protein, transmembrane protein 100 (Tmem100), that is often coexpressed and forms complexes with TRPA1 and TRPV1 in DRG neurons by weakening the TRPA1-TRPV1 association [100,101].

In CD4^+^ T lymphocytes, Bertin et al. [34] described a higher and more sustained level of Ca^2+^ influx upon TcR stimulation in pore-loop domain-deleted TRPA1 harboring KO mice (Trpa1−/−), similarly to what is observed with ionomycin. They experienced no differences in Ca^2+^ influx between WT and Trpa1−/− CD4^+^ T cells following stimulation with the sarcoplasmic reticulum Ca^2+^-ATPase (SERCA) pump inhibitor, thapsigargin, which bypasses proximal TcR signaling and induces SOCE. Moreover, genetic deletion and pharmacological inhibition of TRPV1 in Trpa1−/− CD4^+^ T cells significantly decreased the elevated TcR-gated currents. In our previous work, comparing the pore-loop domain-deleted KO mice to WT, we did not observe significant differences in basal intracellular Ca^2+^ level and in TcR-induced Ca^2+^ signal in T lymphocytes originated from thymus and spleen, but we could detect a significant difference in imiquimode-stimulated sustained elevation of intracellular Ca^2+^ level in CD8^+^ T cells [39]. In cultures of murine spleen T lymphocytes, enriched to 95% purity, live cell Ca^2+^ imaging revealed AITC stimulated elevation of intracellular Ca^2+^ levels, while TRPA1 inhibitors, namely A-967079 as well as HC-030031, reduced AITC-mediated increase in Ca^2+^ level [35]. They also found that TcR-mediated (α-CD3/CD28 bead) stimulation of these cells increased intracellular Ca^2+^ level, while TRPA1 inhibition by A-967079 and HC-030031 significantly reduced this level. To resolve the discrepancies in these experimental results, and to further test the mechanism whether TRPA1 influences TcR dependent Ca^2+^ signaling in T cells, (a) by controlling the driving force for SOCE through modifying plasma membrane potential, (b) through providing Ca^2+^ microdomains or regulating the frequency or amplitude of Ca^2+^ oscillation, or (c) by other mechanisms such as protein-protein interactions, needs further investigation.

Taken together, the T lymphocyte-specific inhibitory effects of JT010, a known specific covalent ligand of TRPA1, on the TcR-stimulated Ca^2+^ signal may be a result of a TRPA-dependent regulation of the T cell activation, but the mechanisms behind it merit further investigation. The question remains whether this effect is caused by: (a) a direct JT010 binding and activation of TRPA1, (b) an alteration of TRPA1 interactions with other TRP channels (e.g., TRPV1), (c) an alteration of TRPA1 interactions with other proteins, or d) a JT010 effect on other JT010-binding proteins. Further experiments with comparisons of WT mice and pore-deficient and full-TRPA KO mice, completed with electrophysiological studies, could answer whether the JT010 inhibitory effect on TcR-dependent Ca^2+^ signal is due to the sustained elevation of intracellular Ca^2+^ level we observed.

## 5. Conclusions

Our results confirmed that TRPA1 does not appear to be a key regulator of the TcR-induced Ca^2+^ signaling in CD4^+^ lymphocytes. The covalent ligand TRPA1 agonist, JT010, inhibits the TcR-induced Ca^2+^ signal in these cells. The significance and pharmacological relevance of these results is to highlight the expression pattern and the physiological function of TRPA1 receptors in the immune cells, which will influence the potential therapeutic use of TRPA1 agonists or antagonists. Further studies are needed to evaluate the effects of the JT010 and other potent TRPA1 ligands in mouse models, and human TRPA1-expressing systems.

## Figures and Tables

**Figure 1 biomolecules-14-00632-f001:**
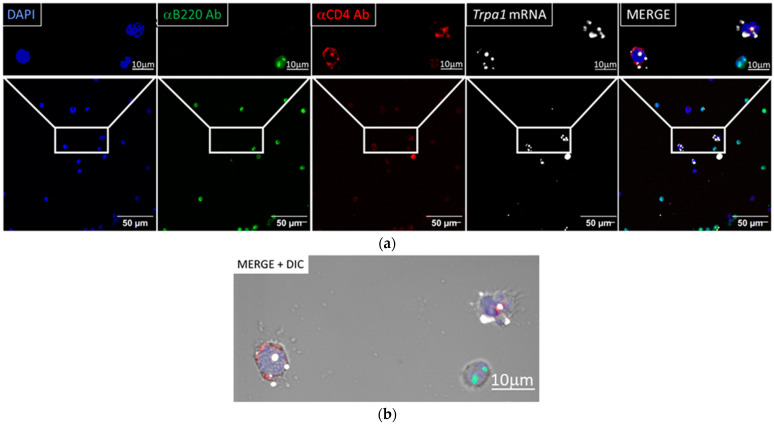
*Trpa1* mRNA expression in CD4^+^ cells. (**a**) Representative fluorescence images showing B220-positive B cells (green, αB220 Ab), CD4-positive cells (red, αCD4 Ab) and their co-localization with the *Trpa1* mRNA transcripts (white) in mouse peritoneal cavity samples. (**b**) The images illustrating the individual channels depicting the identical area are shown merged with a differential interference contrast (DIC) photomicrograph. Nuclear counterstaining was performed with 4′,6- diamidino-2-phenylindole (DAPI, blue).

**Figure 2 biomolecules-14-00632-f002:**
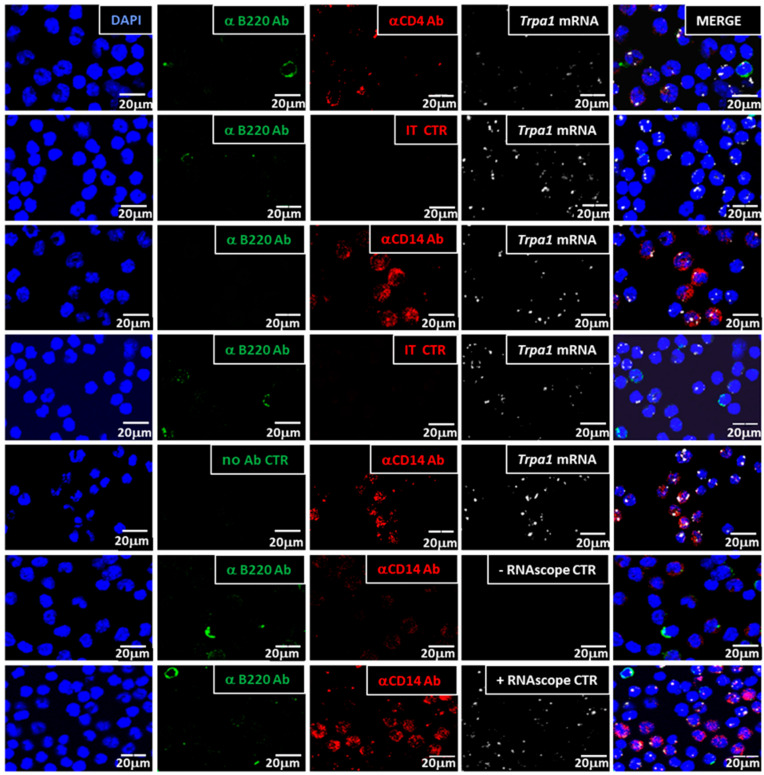
*Trpa1* mRNA co-localization with CD4, CD14, and B220 CD markers. Representative RNAscope results compared to RNAscope positive (+) and negative (–) controls. White RNAscope^®^ ISH signals in the images indicated one of the following: *Trpa1* mRNA, RNA polymerase II subunit A mRNA (+RNA scope CTR), or bacterial dihydrodipicolinate reductase mRNA (-RNA scope CTR). Red immunofluorescent signals in the images indicate one of the following: anti-CD4 antibody (αCD4 Ab), anti-CD14 antibody (αCD14 Ab), or isotype control antibody-labeled (IT CTR) cells. Green immunofluorescent signals in the images indicate the anti-B220 antibody labeled cells (αB220 Ab). Nuclear counterstaining was performed with 4′,6- diamidino-2-phenylindole (DAPI, blue).

**Figure 3 biomolecules-14-00632-f003:**
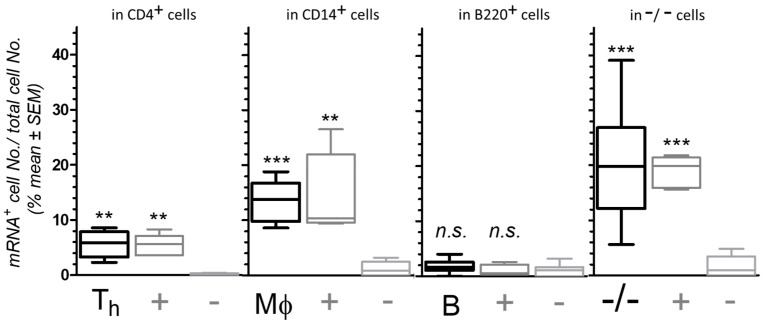
Quantification of *Trpa1* mRNA expression by dual RNAscope^®^ ISH-IF technique. *Trpa1* mRNA expression in CD4^+^ cells (T helper: T_h_), CD14+ (Mϕ), B220^+^ positive (B), and non-CD marker-labeled (−/−) cells were compared to RNAscope positive (+) and negative (−) control signal co-localization in the same cell types, in CD4^+^, CD14^+^, B220^+^ positive and non-CD marker-labeled −/− cells. Number of cells double labeled with both the RNAscope signals and with the particular CD marker antibody was counted, as well as the total cell number based on the DAPI-stained nuclei in the same image. The ratio of the double-labeled cells to the total cell number is expressed as the percentage of mRNA-expressing cells. Statistically significant differences were signed between the values of the *Trpa1* mRNA or the RNAscope + control co-localization compared to co-localization of the RNAscope–control. Data are presented as mean +/− SEM values, n = 5–16, ** *p* < 0.01, *** *p* < 0.01, n.s. non-significant.

**Figure 4 biomolecules-14-00632-f004:**
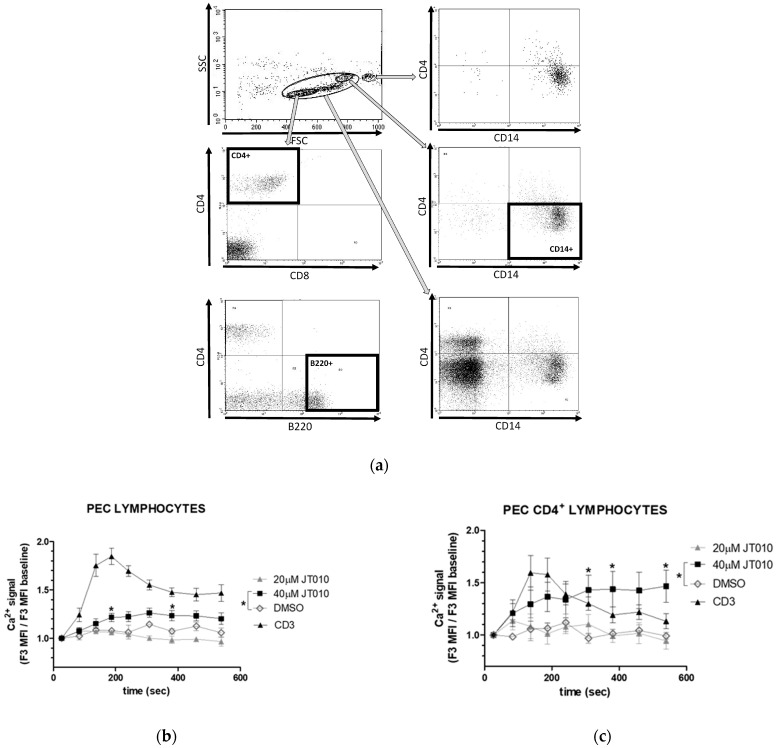
Analyses of concentration-dependent effects of JT010 alone on mice peritoneal (PEC) cells and thymocytes. (**a**) Gating strategy and representative flow cytometry plots of peritoneal cells stained by antibodies to CD4, CD8, B220, and CD14 combinations. Upper left panel shows G1–G4 gates based on FSC versus SSC dot plot of cells. Middle and bottom left panels show cells of G1 gate stained either anti-CD4 and CD8 antibodies or anti-CD4 and B220 antibodies. Right panels show results of anti-CD4 and anti-CD14 labeling of the cells in gates G4, G3, and G2, in order from top to bottom, respectively. Analysis gates surrounded by thick rectangles sign gates for CD4^+^ cells, B220^+^ cells, and CD14^+^ cells in the middle left panel, in the bottom left panel, and in the middle right panel, respectively. Graphs (**b**–**d**) show JT010, DMSO, or TcR (CD3)-stimulated time-dependent changes in intracellular Ca^2+^ levels (**b**) in PEC lymphocytes (G1 gate), (**c**) PEC CD4^+^ lymphocytes, or (**d**) thymocytes. Isolated cells were surface-labeled with CD4 antibody and then loaded with Fluo-3-AM. Fluo-3 fluorescence that is proportional to the intracellular Ca^2+^ levels was detected by flow cytometry. Mean fluorescence intensity (MFI) of non-stimulated cells was measured, then different concentrations of JT010, DMSO as vehicle control-induced changes, and TcR-activated Ca^2+^ signals (CD3) were monitored. Graphs show fold of changes in free intracellular Ca^2+^-levels calculated as a ratio to that of quiescent cells. Data are presented as mean ± SEM, n = 3, * *p* < 0.05. (**e**) Phosphatidydylserine (PS) exposure of cells monitored by fluorescent Annexin V-binding measured by flow cytometry. Addition of 4, 8, 20, or 40 μM of JT010, vehicle control DMSO, and TcR-stimulated (CD3) time-dependent changes in mean fluorescent intensity (MFI) values over time of a representative experiment series are shown in PEC lymphocytes. At the end of each experiment, addition of ionomycin for internal control shows maximum response of PS exposure. Data are presented as mean ± SEM, Figure 4b,c n = 4, Figure 4d n = 3, Figure 4e representative measurement, * *p* < 0.05.

**Figure 5 biomolecules-14-00632-f005:**
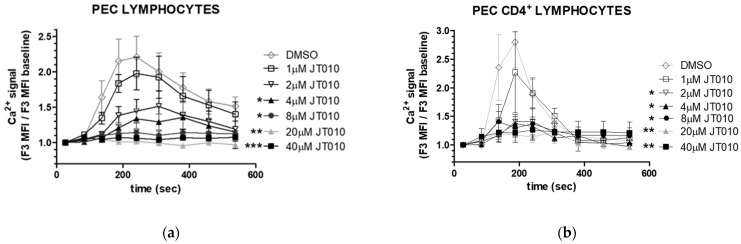
Effects of JT010 on TcR-induced calcium signals and on plasma membrane integrity in lymphocytes and in CD4^+^ lymphocytes isolated from peritoneal cavity of mice. (**a**–**d**) Isolated cells were labeled with anti-CD4 and anti-CD8 antibodies, then incubated with 1–40 μM of JT010 or vehicle control DMSO for 10 min before activation of cells via the T cell receptor (TcR)-induced by anti-CD3 antibody cross-linking. Changes in intracellular Ca^2+^ levels in cells were assessed by monitoring the fluorescence of the Ca^2+^ indicator Fluo-3 by flow cytometry. Lymphocytes (**a**,**c**,**e**) and CD4^+^ lymphocytes (**b**,**d**) were gated as detailed in Figure 4a. (**a**,**b**) Graphs show TcR-stimulated time-dependent changes in intracellular Ca^2+^ levels over time; (**c**,**d**) show JT010 concentration dependence of the maximal response measured at 240 s. Data are presented as mean ± SEM, n = 3, * *p* < 0.05, ** *p* < 0.01, *** *p* < 0.001. (**e**) Effects of 40 μM JT010 on the plasma membrane integrity of lymphocytes measured by annexin V–binding assay. Changes in fluorescent annexin V–binding to cell surface (upper panel), and 7-AminoactinomycinD (7-AAD) labeling of the same cells (middle panel) were recorded by flow cytometry over time. Cell-surface-bound Annexin V fluorescence is proportional to the phosphatidylserine (PS) exposure of the cells and 7-AAD fluorescence is proportional to cell DNA staining in nuclei, both reflecting compromised plasma membrane integrity. At the end of the experiment, ionomycin was applied for internal control to check nonresponsive cells and acquire a maximum response of PS exposure. Bottom left panel shows Annexin V fluorescence plotted against 7-AAD fluorescence at the beginning of the measurement; bottom right panel shows the same after the addition of ionomycin. Representative flow cytometric plots demonstrate the results of n = 3 experiment.

**Figure 6 biomolecules-14-00632-f006:**
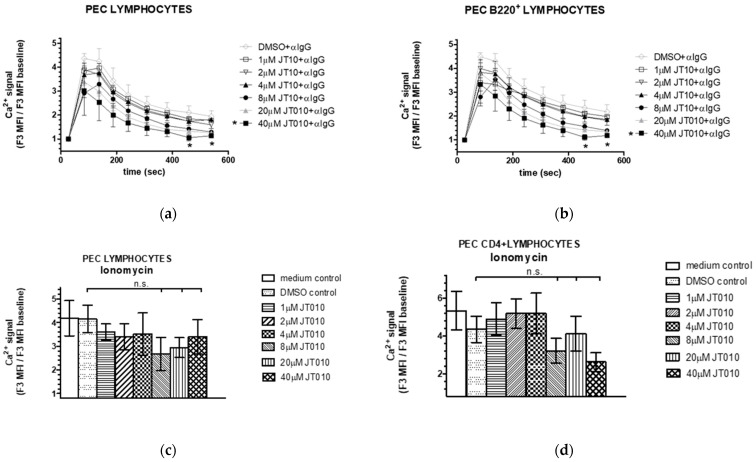
Effect of JT010 on peritoneal B cell activation and ionomycin-stimulated elevation of intracellular Ca^2+^ level of peritoneal lymphocytes. Isolated cells were labeled with anti-B220 and anti-CD4 antibodies, then incubated with 1–40 μM of JT010, vehicle control DMSO, or CaRPMI (medium) for 10 min before activation of cells via the B cell receptor induced by anti-mouse-IgG (**a**,**b**) or 2 μM ionomycin (**a**,**b**). Changes in intracellular Ca^2+^ levels in cells were assessed by monitoring the fluorescence of the Ca^2+^ indicator Fluo-3 by flow cytometry. Lymphocytes (**a**,**c**), B220^+^ lymphocytes (**b**) and CD4^+^ lymphocytes (**d**) were gated as detailed in Figure 4a. Graphs show (**a**,**b**) IgG stimulated time-dependent changes in intracellular Ca^2+^ levels, (**c**,**d**) JT010 concentration dependence of the maximal response by ionomycin measured after 2 min of addition of ionomycin in peritoneal lymphocytes (**c**) or CD4^+^ lymphocytes (**d**). Data are presented as mean ± SEM, n = 3, * *p* < 0.05, n.s. non-significant.

## Data Availability

The original contributions presented in the study are included in the article/Supplementary Material; further inquiries can be directed to the corresponding authors.

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
