# Peer review of "TRPA1 Covalent Ligand JT010 Modifies T Lymphocyte Activation"

_biomolecules, 2024, doi:10.3390/biom14060632_

Round 1

Reviewer 1 Report

Comments and Suggestions for Authors

This paper is of interest as more TRPA1 antagonists are proposed for testing in animal models and human trials. The role of TRPA1 in inflammation is vital and this paper addresses controversy as to the effects of agonists and antagonists and builds upon the work previously published by the laboratory. 

The focus on TRPA1 mRNA is courageous, as it is always found in small volumes, but I wonder why they have not done any analysis to test for protein expression, as well as testing function, and why they have not used the TRPA1 KOs, which they have in their laboratory, to test for the specificity of JT010. This is perhaps just the start of their journey, as they list a number of important experiments to do in the conclusion. It would be good to see if other TRPA1 agonists also have the same effect, or if they can inhibit T cell function or cytokine release etc. 

Overall, they have used appropriate controls and standardisation for planning and analysis. 

Comments on the Quality of English Language

I support the publication of the paper but with several edits to make it easier for the reader to follow. For instance, the following sentence switches between positive and negative wording to describe what is happening, leaving it very difficult to read fluently. This editing is needed throughout. 

 "As the representative plots in Fig. 5e demonstrate, 40 M of JT010 treatment did 433 caused elevation of neither Annexin V nor 7-AAD fluorescence of PEC lymphocytes, only 434 addition of ionomycin (2 M) at the end of the experimentresulted maximum PS exposure 435 of cells. "

Author Response

Response to Comments and Suggestions of Reviewer 1

Thank you for your constructive suggestions and review.

(The lines are numbered as indicated in the Tracked version of the revised Article in our answers,)

  1. Comment and Suggestions 1.1-1.4: “but I wonder why they have not done any analysis to test for protein expression, as well as testing function, and why they have not used the TRPA1 KOs, which they have in their laboratory, to test for the specificity of JT010. This is perhaps just the start of their journey, as they list a number of important experiments to do in the conclusion. It would be good to see if other TRPA1 agonists also have the same effect, or if they can inhibit T cell function or cytokine release etc. “

1.1 Answer: In terms of the suggested analysis to test for protein expression: Our research group investigated the expression of Trpa1 in several brain areas by the ultrasensitive RNAscope in situ hybridization that allows single molecule detection and we found that the central expression of Trpa1 is generally low, but we identified the centrally projecting Edinger-Westphal nucleus (EWcp) where a relatively high Trpa1 mRNA expression was found, but restricted to the urocortin 1 (UCN1)-expressing peptidergic cells(Konkoly et al., 2021; Oláh et al., 2021; Kormos et al., 2022; Konkoly et al., 2022; Al-Omari et al., 2023). We applied this method here too, because the specificity of the commercially available TRPA1 antibodies is at least questionable (Virk et al., 2019; Patil et al., 2023; Royas-Galvan et al., 2024) and none of them appeared to be specific in our hands using immunohistochemistry or immunfluorescent labeling of the cells visualized by confocal microscopy.

1.2Answer: In terms of the suggested analysis testing function: We most recently examined the electrophysiological features of EWcp/UCN1 cells in acute brain slices, to test whether TRPA1 is functionally active on EWcp neurons by patch clamp recordings in whole cell configuration (Al-Omari et al., 2023). We have selected this cell population because they express Trpa1 mRNA in a relatively high amount (see above also our answer No. 1.1.) Application of JT010, a selective and potent TRPA1 agonist, significantly increased the spontaneous firing frequency of UCN1-immunoreactive neurons while it was ineffective in neighboring neurons lacking Trpa1 and UCN1. To our knowledge, this is the first evidence suggesting the functional role of TRPA1 in neurons of the mouse brain. Thank you for your suggestion on electrophysiological experiments by determining effects of JT010 in immune cells. We augmented now the importance of this approach in the Discussion lines 645-646. in the row of the listed further experiments.

1.3Answer: In terms of the suggested analysis using the TRPA1 KOs to test for the specificity of JT010: As we wrote in the Discussion lines 644-647, further experiments with comparison of pore-deficient KO mice, that we have in our laboratory, and full-TRPA KO mice and WT mice could answer if the JT010 inhibitory effect on TcR-dependent Ca2+ signal is due to the TRPA1 dependent sustained elevation of intracellular Ca2+ level we observed. In our previous study, our findings that specific protein staining was detected both in mononuclear cells of WT and KO mice suggested that at least the N-terminal part of the protein was expressed in monocytes, CD4+ and CD8+ lymphocytes both in WT and pore-loop deleted KO mice. Here we compared immunofluorescence labeling of immune cells by an anti-TRPA1 antibody that recognizes the first 100 amino acids of the channel. Our results also indicated the presence of a truncated transcript of TRPA1 in immune cells, similarly as reported in Supplemental Data Fig.S1 by Bautista et. al (2013) for TRG tissue of the same strain of TRPA1 functional deficient KO mice. The important JT010 binding cysteines (C622, C666 in mouse) and another residues of the JT010 binding pocket located in the N terminal of the protein, this way the absence of JT010 inhibitory effect on TcR-dependent Ca2+ signal could only prove that the mechanism by which JT010 inhibits TcR-dependent Ca2+ signal is due to the TRPA1 dependent changes in intracellular Ca2+ level, not the TRPA1 specificity of JT010. In accordance with your comment, our present study seems to be only a start of a journey.

1.4Answer: In terms of the suggested analysis using other TRPA1 agonists also have the same effect, or if they can inhibit T cell function or cytokine release: As reported earlier in our previous studies (Szabó et al 2022), addition of agonists (such as AITC or cinnamaldehyde) either had no effect on TcR dependent Ca2+ signal at lower concentrations or caused cell shrinkage (measured by flow cytometry FSC values), granulation (measured by flow cytometry SSC values), and compromised membrane permeability (shown by PI positive staining of the cells) at higher concentrations in immune cells in our system. Similar damage of these primary immune cells have been observed by these agents in the attempted cytokine release experiments. Similarly, inhibitors at concentrations that have been used previously successfully blocking mouse TRPA1 function in CHO, HEK and other cell types (50-100 μM of HC-030031, 0.5-1 μM A967079), cell shrinkage, granulation and compromised membrane permeability occurred, indicating some type of (most probably) necrotic cell death. Lower concentrations of antagonist (10-20 μM of HC, 0.1-0.2 μM A96) did not modify TcR dependent Ca2+ signal of the T cells.

  1. Comment and Suggestion:.”I support the publication of the paper but with several edits to make it easier for the reader to follow. For instance, the following sentence switches between positive and negative wording to describe what is happening, leaving it very difficult to read fluently. This editing is needed throughout.
    • 2. 1 Answer: The sentence in lines 442-445 has been corrected. "As the representative plots in Fig. 5e demonstrate, 40 mM of JT010 treatment did not cause elevation of the Annexin V labeling or the 7-AAD fluorescence of PEC lymphocytes, only addition of ionomycin (2 mM) at the end of the experiment resulted maximum PS exposure of the cells."
    • 2.2 Answer: Similar overcomplicated sentences have been corrected also throughout of the manuscript by editing by an experienced English-speaking colleague.

  1. Suggestion the manuscript requires moderate English revisions”

 3.Answer: The manuscript has been reviewed by an experienced English-speaking colleague living in the US (as indicated in the acknowledgement).

Virk HS, Rekas MZ, Biddle MS, Wright AKA, Sousa J, Weston CA, Chachi L, Roach KM, Bradding P. Validation of antibodies for the specific detection of human TRPA1. Sci Rep. 2019 Dec 6;9(1):18500. doi: 10.1038/s41598-019-55133-7. PMID: 31811235; PMCID: PMC6898672.

Rojas-Galvan NS, Ciotu CI, Heber S, Fischer MJM. Correlation of TRPA1 RNAscope and Agonist Responses. J Histochem Cytochem. 2024 May;72(5):275-287. doi: 10.1369/00221554241251904. Epub 2024 May 10. PMID: 38725415.

Patil MJ, Kim SH, Bahia PK, Nair SS, Darcey TS, Fiallo J, Zhu XX, Frisina RD, Hadley SH, Taylor-Clark TE. A Novel Flp Reporter Mouse Shows That TRPA1 Expression Is Largely Limited to Sensory Neuron Subsets. eNeuro. 2023 Dec 4;10(12):ENEURO.0350-23.2023. doi: 10.1523/ENEURO.0350-23.2023. PMID: 37989590; PMCID: PMC10698635.

Bautista, D.M.; Pellegrino, M.; Tsunozaki, M. TRPA1: A gatekeeper for inflammation. Annu Rev Physiol 2013, 75, 181-200, doi:10.1146/annurev-physiol-030212-183811.

Reviewer 2 Report

Comments and Suggestions for Authors

General comments:

Szabo et al. analyze the expression and function of TRPA1 in immune cells derived from the peritoneal cavity of mice. The authors show that CD4+ T cells express a low, nonetheless  significant expressing of TRPA1 mRNA and display concentration-dependent suppression of TcR/Ca2+ entry-triggered cell activation in response to JT010, a covalent TRPA1 agonist. It is shown that B-cells lack both detectable mRNA expression and comparable inhibitory responses to JT010. Unfortunately, there is a recent publication in IJMS (Matsubara et al.) reporting compelling evidence against reasonable ability of JT010 modify /activate murine TRPA1. This appears as a serious problem as the used tools appears inappropriate or at least barely useful. I would ask the authors to comment and resolve this issue, if possible. In case there is no solid evidence for suitability of their pharmacological strategy, I so not see significance of their study. Moreover, it is obvious that the authors can, to this end, not provide reasonable information on the mechanism by which JT010, at the rather high concentrations used, inhibits T-cell activation. There is very little discussion about the plausible, potential mechanisms. The findings of this study raise a number of essential questions, which need to be addressed.

Specific comments and suggestions:

1)      It is critical that the authors explain, why they think JT010 is a suitable tool to investigate TRPA1 in murine immune cells in view of the recently published paper by Matsubara et al. (https://www.mdpi.com/1422-0067/23/22/14297). Why have the authors not chosen other tools (inhibtors/activators) that more convincingly target murine TRPA1?

2)      The results of this study may be interpreted in terms of a JT010-induced (potentially TRPA1-mediated) suppression of T-cell activation by interference with Ca2+ handling. In the discussion the authors briefly mention that cation entry via TRPA1 may reduce the driving force for Ca2+ entry associated with the expected membrane depolarization. In addition, several alternative, potential mechanisms are listed including even alternative JT010 targets, which appear likely at the used concentrations. I think, an impact of JT010 on membrane potential is certainly plausible considering SOCE, which is remarkably dependent on cell polarization, as the relevant Ca2+ entry pathway in T-cells. This concept could be scrutinized rather easily by determining effects of JT010 on membrane potential. It would be highly valuable to clarify this issue or, at least to extend the discussion on this topic.

3)      Receptor-triggered Ca2+ signaling differs remarkably between T- and B-cells. Hence, the lack of JT010 effects in B-cell does not necessarily prove a lack of functional TRPA1 (or of an alternative JT010 target) in the B-cells. It would be informative to see a comparison of JT010 effects in B- and T-cells activated by a more comparable activation scenario such as ER depletion by thapsigargin. Differences in T- and B-cell Ca2+ signaling may deserve some attention in the discussion.

4)       The authors analyze expression of TRPA1 at mRNA level. This leaves a key question open: to what degree is the channel protein expressed and targeted to the plasma membrane of T-cells. Is there any solid evidence for TRPA1 expression and plasma membrane targeting in T-cells or other immune cells from high resolution fluorescence microscopy or electrophysiology? Please discuss.

Minor point: In their brief discussion of functional aspects the authors mention the potential control of SOCE by TRPM4 via changes in membrane potential and generation of Ca2+ microdomains. This statement may not be immediately conceivable for readers outside the field and original references are lacking. Similarly, in line 564…”TRPC1, TRPC3, TRPV1 expression and were also reported…”  This sentence is unclear and requires appropriate original reports cited. I strongly suggest to remodel this section.

Comments on the Quality of English Language

The manuscript is well written. Nonetheless, moderate language editing and amendment of wording in some parts is required.

Author Response

Response to Comments and Suggestions of Reviewer 2

(The lines are numbered as indicated in the Tracked version of the corrected Article.)

Thank you for your general comments, for the thorough review, and for the constructive and substantial suggestions.

1.1 General comment:

Unfortunately, there is a recent publication in IJMS (Matsubara et al.) reporting compelling evidence against reasonable ability of JT010 modify /activate murine TRPA1. This appears as a serious problem as the used tools appears inappropriate or at least barely useful. I would ask the authors to comment and resolve this issue, if possible. In case there is no solid evidence for suitability of their pharmacological strategy, I so not see significance of their study.”

1.1 Answer to General comment: We referred Matsubara et al. results (2022) in the earlier version of the manuscript, in the Results section 3.2. and in the Discussion, Reference 70, in lines 384-387 and in lines 558-562, Reference 45, in lines 402-405, and lines 564-567 in the revised manuscript.

According to your suggestion, we have now supplemented the Introduction lines 72-91 and the Discussion with lines 568-582 in the revised version of the manuscript:

Introduction lines 72-91:

Matsubara et al [45] provided evidence that JT010 is a much weaker TRPA1 agonist in mice dorsal root ganglion cells than in human fibroblast-like synoviocytes, at concentrations under 100 nM. They also reported that JT010 is a potent activator of human TRPA1 (hTRPA1), but not mouse TRPA1 (mTRPA1) expressed in human embryonic kidney (HEK) cells. However, since two critical cysteines, C621 (important for JT010 interaction) and C665 of hTRPA1 are conserved in mTRPA1 (C622 and C666) [46-48], we intended to test if application of higher concentrations of JT010 modulates TRPA1-mediated Ca2+ influx in murine peritoneal cells. We aimed to test the effect of JT010 on mouse T cell function because of two reasons: 1) previous functional results indicated differences in calcium signaling and in cytokine secretion between WT and TRPA1 KO mice [34, 39], and 2) in contrast to AITC and cinnamaldehyde, JT010 does not cause membrane damage of the primary immune cells even at high concentrations. In our earlier work [39], we could not determine the influence on TRPA1-mediated Ca2+ influx by other agonists in splenocytes and thymocytes, because these primary immune cells were vulnerable to these agents in our system, indicated by cell shrinkage, granulation, and compromised membrane permeability (shown by propidium iodide positive staining of the cells). We could not detect any significant differences in the TcR-induced Ca2+ signal of T lymphocytes isolated from spleen and thymus between the WT and pore-loop domain deficient TRPA1 KO mice. However, imiquimod-stimulated elevation of Ca2+ level with sustained time kinetics was higher in the mouse TRPA1 KO CD8+ thymocytes [39].

Discussion with lines 568-582:

The half maximum concentration required for hTRPA1 activation is reported to be 0.65 nM in calcium uptake assays and 7.6 nM in electrophysiological studies [43, 48]. Matsubara et al. estimated the half maximum concentration for hTRPA1 activation to be 3-10 nM, while application of higher JT010 concentrations (at maximum 1000nM) induced a moderate Ca2+ response to mTRPA1 activation suggesting that mTRPA1 is much less sensitive to JT010 than hTRPA1. The highest concentration of JT010 applied in electrophysiological experiments were 100nM of JT010. The two critical cysteines, C621 (important for JT010 interaction) and C665 of hTRPA1 are conserved in mTRPA1 (C622 and C666) [46-48], Matsubara et al identified non-conserved residues in hTRPA1 and mTRPA1 that influenced the response to JT010, by measuring the ratio of the peak Ca2+ response of mutant TRPA1 versions compared to control HEK cells and in electrophysiological experiments. They found that even if the different residues of mTRPA1 were substituted with cysteine of hTRPA1, these mutants showed low sensitivity to 100nM JT010 in patch clamp experiments. This suggests that the much weaker potency of JT010 to activate TRPA1 reflects a more complex determination of TRPA1 interaction with JT010.

1.2 General comment:

“Moreover, it is obvious that the authors can, to this end, not provide reasonable information on the mechanism by which JT010, at the rather high concentrations used, inhibits T-cell activation. There is very little discussion about the plausible, potential mechanisms. The findings of this study raise a number of essential questions, which need to be addressed.”

1.2 Answer to General comment: We agree with the reviewer that, based on our previous and present results, to this end, we can not provide reasonable information on the mechanism by which JT010 inhibits T-cell activation. This was the reason while we only mentioned some potential mechanisms in the Discussion (lines 560-566), and listed a number of important experiments to do to address the mechanism(s) behind it in the Discussion (lines 570-575) in the earlier version of the manuscript.

However, according to your comment that “the findings of this study raise a number of essential questions which need to be addressed”, we have now supplemented the Discussion with lines 583-637:

 Several TRP channels have been reported as important regulators of TcR signaling [74], by regulating the activity of store-operated Ca2+ signaling (SOCE), which is dependent on cell membrane depolarization [86]. TRPM4, a Ca2+-activated Na+ selective channel in T cells serves as one of the negative feedback mechanism during T cell activation upon SOCE [87-88]. Ca2+-induced activation of TRPM4 triggers Na+ influx through the channel, it depolarize the plasma membrane and restrict further Ca2+ influx through ORA1.) The Mg2+- and Ca2+- permeable TRPM7 by its kinase domain activity regulates ORAI1 signaling and coordinates antigen receptor signaling termination in B lymphocytes, likely through phosphorylation of phospholipase Cγ (PLCγ) isoforms1 [89]. Knockdown of TRPC3 caused a small reduction in SOCE [90], function of SOCE in human platelets required the association of STIM1, the Ca2 +-permeable channel Orai1, TRPC1 and TRPC6 [91-93]. A potential mechanism for TRPA1 dependent modulation of SOCE was also proposed in megakaryocytes, a protein-protein binding transmitted association of TRPA1 with STIM1, Orai1, TRPC1 and TRPC6, based on the results of thrombin stimulated co-immunoprecipitation of these proteins [94]. One of the characteristic feature of TRPA1 is the high number of ankyrin repeats at the cytoplasmic NH2-terminal tail, which might provide the protein elasticity, as well as, the ability to interact with other proteins [1]. The interactions of TRPA1 channel with other proteins is an important research area, most probably in terms of regulation of both function and intracellular localization, trafficking, as well as internalization of TRPA1 [1]. The best studied example is so far TRPA1 and TRPV1, where coexpression of the channels results distinct activation profiles from those of cells expressing only TRPA1 or TRPV1 [95-97]. In neuronal cells, TRPA1 is generally co-expressed with TRPV1, and in vitro direct evidence was reported that formation of a channel by two molecules of TRPA1 together with two monomers of TRPV1 is possible [98]. A mutation in TRPV1 influenced the voltage dependency, Ca2+ sensitivity and magnitude of TRPA1 mediated currents [99]. The Ca2+ -triggered activation of TRPA1 is attenuated by TRPV1 in the presence of extracellular Ca2+, but not in Ca2+-free conditions, suggesting that this modulation may work rather as a direct protein-protein interaction then through TRPV1 providing Ca2+ microdomains for TRPA1 activation. The interaction may also act through another coupling protein, transmembrane protein 100 (Tmem100) that is often coexpressed and forms complexes with TRPA1 and TRPV1 in DRG neurons, by weakening the TRPA1-TRPV1 association [100-101].

In CD4+ T lymphocytes, Bertin et al [34] described a higher and more sustained level of Ca2+ influx upon TcR stimulation in pore-loop domain deleted TRPA1 harboring KO mice (Trpa1−/−), similarly to what observed with ionomycin. They experienced no differences in Ca2+ influx between WT and Trpa1−/− CD4+ T cells following stimulation with the sarcoplasmic reticulum Ca2+-ATPase (SERCA) pump inhibitor, thapsigargin, which bypasses proximal TcR signaling and induces SOCE. Moreover, genetic deletion and pharmacological inhibition of TRPV1 in Trpa1−/− CD4+ T cells significantly decreased the elevated TcR-gated currents. In our previous work, comparing the pore-loop domain deleted KO mice to WT we did not observed significant differences in basal intracellular Ca2+ level and in TcR-induced Ca2+ signal in T lymphocytes originated from thymus and spleen, but we could detect significant difference in imiquimode stimulated sustained elevation of intracellular Ca2+ level in CD8+ T cells [39]. In cultures of murine spleen T lymphocytes, enriched to 95% purity, live cell Ca2+ imaging revealed AITC stimulated elevation of intracellular Ca2+ levels, while TRPA1 inhibitors, namely A-967079 as well as HC-030031, reduced AITC-mediated increase in Ca2+ level [35]. They also found that TcR-mediated (α-CD3/CD28 bead) stimulation of these cells increased intracellular Ca2+ level, while TRPA1 inhibition by A-967079 and HC-030031 significantly reduced this elevation. To resolve the discrepancies in these experimental results, and to further test the mechanism whether TRPA1 influences TcR dependent Ca2+ signaling in T cells: a) by controlling the driving force for SOCE through modifying plasma membrane potential, b) through providing Ca2+ microdomains or regulating the frequency or amplitude of Ca2+ oscillation, or c) by other mechanisms such as protein-protein interactions, needs further investigations.

As pointed out in the Abstract in lines 20-22, the aim of this study was only to address two controversial aspects of TRPA1 in lymphocytes, built upon our work we previously published: 1) further elucidate the expression of Trpa1 mRNA in immune cells by RNAscope in situ hybridization and 2) further test the role of TRPA1 in lymphocyte activation without enrichment, sorting or separation methods, with minimal intervention of cell activation or stage, by using an agent that not influence cell membrane integrity.

 Specific comment and Suggestion

  1. “It is critical that the authors explain, why they think JT010 is a suitable tool to investigate TRPA1 in murine immune cells in view of the recently published paper by Matsubara et al. (https://www.mdpi.com/1422-0067/23/22/14297). Why have the authors not chosen other tools (inhibtors/activators) that more convincingly target murine TRPA1?
  2. Answer to Specific comment and Suggestion 1.

In terms of using JT010 in murine immune cells, please read 1.1 Answer to General comment: above. As detailed in the answer of 1.1 General comment, we supplemented the Introduction the Introduction lines 72-91 and the Discussion with lines 568-582 in the revised version of the manuscript.

In addition, as reported earlier (Szabó et al 2022), the reason why we did not choose other inhibitors/activators that more convincingly target murine TRPA1 is, that we tried several of them in primary immune cells. At concentrations that have been used previously successfully blocking mice TRPA1 function in CHO, HEK and other cells (50-100 μM of HC-030031, 0.5-1 μM A967079), cell schrinkage (measured by flow cytometry FSC values), granulation (measured by flow cytometry SSC values), and compromised membrane permeability (shown by PI positive staining of the cells) occurred, indicating some type of (most probably) necrotic cell death. Lower concentrations of antagonist (10-20 μM of HC030031, 0.1-0.2 μM A967079) did not modify TcR dependent Ca2+ signal of the T cells. Addition of agonists or bimodal modulators, similarly, either had no effect on TcR dependent Ca2+ signal at lower concentrations, or caused cell shrinkage and compromised membrane at higher concentrations in this immune cells. Similar damage of these primary immune cells have been observed by these agents in the attempted cytokine release experiments.

  1. “The results of this study may be interpreted in terms of a JT010-induced (potentially TRPA1-mediated) suppression of T-cell activation by interference with Ca2+ In the discussion the authors briefly mention that cation entry via TRPA1 may reduce the driving force for Ca2+ entry associated with the expected membrane depolarization. In addition, several alternative, potential mechanisms are listed including even alternative JT010 targets, which appear likely at the used concentrations. I think, an impact of JT010 on membrane potential is certainly plausible considering SOCE, which is remarkably dependent on cell polarization, as the relevant Ca2+ entry pathway in T-cells. This concept could be scrutinized rather easily by determining effects of JT010 on membrane potential. It would be highly valuable to clarify this issue or, at least to extend the discussion on this topic.

2.1 Answer to Specific comment and Suggestion 2.:

According to your suggestion to extend the Discussion on this topic, we supplemented the Discussion with lines 583-637 in the revised version of the manuscript.

2.2 Answer to Specific comment and Suggestion 2.:

Thank you for your suggestion on electrophysiological experiments by determining effects of JT010 on membrane potential. We augmented now the importance of this approach in the Discussion lines 644-646 in the row of the listed further experiments. In terms of the suggested experiments, please also read 1.2 Answer to General comment above.

  1. „Receptor-triggered Ca2+ signaling differs remarkably between T- and B-cells. Hence, the lack of JT010 effects in B-cell does not necessarily prove a lack of functional TRPA1 (or of an alternative JT010 target) in the B-cells. It would be informative to see a comparison of JT010 effects in B- and T-cells activated by a more comparable activation scenario such as ER depletion by thapsigargin. Differences in T- and B-cell Ca2+ signaling may deserve some attention in the discussion.”
  2. Answer to Specific comment and Suggestion 3.

Our main questions of this work were only to address two controversial issues on TRPA1 presence and function in immune cells with minimal intervention of activation or stage of the cells: the debated presence of Trpa1 mRNA in mouse T lymphocytes, and whether TRPA1 function contributes to the activation of T lymphocytes. We agree with the reviewer, that our findings that Trpa1 mRNA expression was not significant in B cells isolated from peritoneal cavity of mice, and that JT010 did not modified elevation of intracellular Ca2+ level of peritoneal B cell activated by anti-IgG antibody, does not prove the lack of functional TRPA1 (or of an alternative JT010 target) in the B-cells. B and T cell activation are indeed fundamentally different processes, even if changes in intracellular Ca2+ levels play an important role in both.

We probed the effect of JT010 on ionophore ionomycin stimulated elevation of intracellular Ca2+ level of cells, as well as, on anti-IgG antibody stimulated activation of B lymphocytes to test if the inhibitory effect of JT010 was restricted to T cell activation in our experimental conditions. A detailed comparison of JT010 effects in B- and T-cells from different sources, by other activation scenarios would be an interesting aim of further studies.

In agreement with your suggestion, we included in the revised Discussion lines 617-620 Bertin et al (2017) results with thapsigargin on Ca2+ influx upon TcR stimulation.

  1. “The authors analyze expression of TRPA1 at mRNA level. This leaves a key question open: to what degree is the channel protein expressed and targeted to the plasma membrane of T-cells. Is there any solid evidence for TRPA1 expression and plasma membrane targeting in T-cells or other immune cells from high resolution fluorescence microscopy or electrophysiology? Please discuss.”

4.Answer to Specific comment and Suggestion 4.:

One of the major challenges in the analyses of TRPA1-related function is the lack of sufficiently specific and commercially available antibodies (Virk et al., 2019; Patil et al., 2023; Royas-Galvan et al., 2024). Although to study desensitization, intracellular trafficking or the fate of the receptor/channel protein would be a crucial point for understanding its role not only in immune cells. Unfortunately, even though we tried different approaches, our efforts failed to develope antibodies recognizing extracellular epitopes-of TRPA1. Therefore, in our previous article, we compared intracellular immunofluorescence labeling of immune cells by an anti-TRPA1 antibody that recognizes the first 100 amino acids of the channel, compared to labeling by its isotype control counterpart, using flow cytometry. The specificity of the anti-TRPA1 antibody was detected as a clear difference in fluorescence compared to staining with isotype control rabbit IgG1 antibody labelled fluorescence. This method revealed that specific staining can be detected in mononuclear cells, in CD4+ and CD8+ lymphocytes, suggesting endogenous TRPA1 protein expression in monocytes, CD4+ and CD8+ lymphocytes. Unfortunately, these results did not prove plasma membrane targeting of TRPA1 protein. The specificity of the commercially available TRPA1 antibodies is at least questionable, none of them appeared to be specific in our hands using immunohistochemistry or immunofluorescent labeling of the cells visualized by confocal microscopy.

While specific available antibodies still missing for TRPA1 detection, Qiao et al (2020) designed, synthesized a “turn on” fluorescent probe for monitoring TRPA1 protein expression in the plasma membrane of live cells. They evaluated the specificity and sensitivity of this photochromic ligand based fluorescent probe by electrophysiology and confocal imaging. This probe displayed higher affinity and selectivity to TRPA1 channel versus all other ion channels tested including TRPV1, TRPV3, Nav1.4, Nav1.5, and hERG. In future experiments, this or similar photochromic ligand based fluorescent probe would be a useful tool to analyse TRPA1 plasma membrane targeting in T-cells or in other immune cells by high resolution fluorescence microscopy.

In terms of electrophysiological assays, Bertin et al (2017) demonstrated TRPA1 channel functionality in cell-attached voltage clamp in CD4+ T cells, both by AITC stimulated whole-cell currents and single-channel recordings.

  1. Minor point: In their brief discussion of functional aspects the authors mention the potential control of SOCE by TRPM4 via changes in membrane potential and generation of Ca2+ This statement may not be immediately conceivable for readers outside the field and original references are lacking. Similarly, in line 564…”TRPC1, TRPC3, TRPV1 expression and were also reported…”  This sentence is unclear and requires citation of appropriate original reports. I strongly suggest to remodel this section.”

5.Answer to Specific comment and Suggestion 5.:

We have now remodelled this section and supplemented the Discussion with lines 583-637.

  1. Suggestion The manuscript is well written. Nonetheless, moderate language editing and amendment of wording in some parts is required.”

6.Answer to Specific comment and Suggestion 6: The manuscript has been reviewed by an experienced English-speaking colleague living in the US (as indicated in the acknowledgement).

Qiao Z, Qi H, Zhang H, Zhou Q, Wei N, Zhang Y, Wang K. Visualizing TRPA1 in the Plasma Membrane for Rapidly Screening Optical Control Agonists via a Photochromic Ligand Based Fluorescent Probe. Anal Chem. 2020 Jan 21;92(2):1934-1939. doi: 10.1021/acs.analchem.9b04193. Epub 2019 Dec 31. PMID: 31855414

Virk HS, Rekas MZ, Biddle MS, Wright AKA, Sousa J, Weston CA, Chachi L, Roach KM, Bradding P. Validation of antibodies for the specific detection of human TRPA1. Sci Rep. 2019 Dec 6;9(1):18500. doi: 10.1038/s41598-019-55133-7. PMID: 31811235; PMCID: PMC6898672.

Rojas-Galvan NS, Ciotu CI, Heber S, Fischer MJM. Correlation of TRPA1 RNAscope and Agonist Responses. J Histochem Cytochem. 2024 May;72(5):275-287. doi: 10.1369/00221554241251904. Epub 2024 May 10. PMID: 38725415.

Patil MJ, Kim SH, Bahia PK, Nair SS, Darcey TS, Fiallo J, Zhu XX, Frisina RD, Hadley SH, Taylor-Clark TE. A Novel Flp Reporter Mouse Shows That TRPA1 Expression Is Largely Limited to Sensory Neuron Subsets. eNeuro. 2023 Dec 4;10(12):ENEURO.0350-23.2023. doi: 10.1523/ENEURO.0350-23.2023. PMID: 37989590; PMCID: PMC10698635.

Round 2

Reviewer 2 Report

Comments and Suggestions for Authors

The authors have done very well with providing a plausible reasoning for use of an apparently problematic pharmacological tool. The results are certainly interesting and valuable although I still consider the pharmacological strategy as all but perfect.

Comments on the Quality of English Language

The manuscript is well written and needs only minor style editing.